# Impacts of Deoxygenation and Hypoxia on Shark Embryos Anti-Predator Behavior and Oxidative Stress

**DOI:** 10.3390/biology12040577

**Published:** 2023-04-10

**Authors:** Jaquelino Varela, Sandra Martins, Melanie Court, Catarina Pereira Santos, José Ricardo Paula, Inês João Ferreira, Mário Diniz, Tiago Repolho, Rui Rosa

**Affiliations:** 1MARE—Marine and Environmetal Sciences Centre/ARNET—Aquatic Research Network, Laboratório Marítimo da Guia, Faculdade de Ciências, Universidade de Lisboa, 2750-374 Cascais, Portugal; 2Sphyrna Association, Sal Rei 5110, Boa Vista Island, Cape Verde; 3Comparative Molecular and Integrative Biology, Centro de Ciências do Mar, Universidade do Algarve, 8005-139 Faro, Portugal; 4Environmental Economics Knowledge Center, Nova School of Business and Economics, New University of Lisbon, 2775-405 Carcavelos, Portugal; 5Departamento de Biologia Animal, Faculdade de Ciências, Universidade de Lisboa, 2750-374 Cascais, Portugal; 6UCIBIO—Applied Molecular Biosciences Unit, Department of Chemistry, School of Science and Technology, NOVA University Lisbon, 2819-516 Caparica, Portugal; 7Associate Laboratory i4HB, Institute for Health and Bioeconomy, School of Science and Technology, NOVA University Lisbon, 2819-516 Caparica, Portugal; 8LAQV—Associated Laboratory for Green Chemistry—REQUIMTE, Department of Chemistry, School of Science and Technology, NOVA University Lisbon, 2829-516 Caparica, Portugal

**Keywords:** climate change, oxygen loss, elasmobranch, embryogenesis, predation, sharks

## Abstract

**Simple Summary:**

Despite their importance, sharks are among the most endangered ocean species. In addition to overexploitation and the destruction of their natural habitat, climate change is also known to pose a serious threat to them. Among the physico-chemical changes associated with climate change, oxygen loss has been the least studied in terms of its effect on shark physiology and behavior. In this study, we evaluated the impact of deoxygenation (93% air saturation) and hypoxia (26% air saturation) on the anti-predatory behavior and physiology of temperate shark embryos. We found that hypoxia caused a high mortality (44%), significantly increased embryo movement within capsules, and, consequently, reduced the freezing response behavior (a behavior that allows embryos to be unnoticed by predators). Regarding oxidative stress, most biomarkers analyzed were not impacted by the experimental treatments. Overall, our results suggest that the temperate shark’s early life stages showed a certain degree of resilience to deoxygenation but not to hypoxia.

**Abstract:**

Climate change is leading to the loss of oxygen content in the oceans and endangering the survival of many marine species. Due to sea surface temperature warming and changing circulation, the ocean has become more stratified and is consequently losing its oxygen content. Oviparous elasmobranchs are particularly vulnerable as they lay their eggs in coastal and shallow areas, where they experience significant oscillations in oxygen levels. Here, we investigated the effects of deoxygenation (93% air saturation) and hypoxia (26% air saturation) during a short-term period (six days) on the anti-predator avoidance behavior and physiology (oxidative stress) of small-spotted catshark (*Scyliorhinus canicula*) embryos. Their survival rate decreased to 88% and 56% under deoxygenation and hypoxia, respectively. The tail beat rates were significantly enhanced in the embryos under hypoxia compared to those exposed to deoxygenation and control conditions, and the freeze response duration showed a significant opposite trend. Yet, at the physiological level, through the analyses of key biomarkers (SOD, CAT, GPx, and GST activities as well as HSP70, Ubiquitin, and MDA levels), we found no evidence of increased oxidative stress and cell damage under hypoxia. Thus, the present findings show that the projected end-of-the-century deoxygenation levels elicit neglectable biological effects on shark embryos. On the other hand, hypoxia causes a high embryo mortality rate. Additionally, hypoxia makes embryos more vulnerable to predators, because the increased tail beat frequency will enhance the release of chemical and physical cues that can be detected by predators. The shortening of the shark freeze response under hypoxia also makes the embryos more prone to predation.

## 1. Introduction

Dissolved oxygen (DO) is one of the most critical limiting factors in the marine environment [1,2]. Yet, oxygen loss is the least studied among the main symptoms of climate change in the oceans [3,4,5]. In the last five decades, the amount of dissolved oxygen in the oceans has dropped by two percent, and models predict a decline of one to seven percent by the end of this century [6,7]. Known as ocean deoxygenation, this phenomenon is linked to global warming, enhanced coastal eutrophication processes, and the expansion of oxygen minimum zones (OMZs) in the pelagic realm [2,6,8]. A reduction in oxygen below the required level causes severe physiological and behavioral changes in marine organisms [9]. Oxygen is an essential element for aerobic organisms to obtain energy from organic matter through cellular respiration [10]. Yet, during this process, reactive oxygen species (ROS) are formed, which include free radicals and their non-radical intermediates, such as the superoxide radical (O_2_^−^), hydrogen peroxide (H_2_O_2_), and the hydroxyl radical (OH) [10,11]. Free radicals are unstable and very reactive atoms or molecules containing unpaired valence shells constantly produced by aerobic organisms during normal cell metabolism [12]. Due to their properties, ROS react readily with lipids, proteins, and DNA, causing cellular damage [13]. To cope with this, organisms prevent or produce antioxidant defense systems that combat the harmful effect of ROS [12,13]. The primary defense mechanism consists of an enzymatic system that promotes the conversion of O_2_^−^ by superoxide dismutase (SOD) into H_2_O_2_ [13,14]. Then catalase (CAT) and glutathione peroxidase (GPx) both act on the conversion of H_2_O_2_ into H_2_O and O_2_ [14,15]. Glutathione-s-transferase (GST) also acts as the first line of defense in detoxification by promoting the elimination of xenobiotics [16]. As a second line of defense, organisms also produce heat shock protein (HSP), which is involved in repairing and refolding denatured proteins [17]. When this system fails, ubiquitin eliminates irreversibly damaged proteins [18]. Oxidative stress occurs when the production of ROS exceeds an organism’s capacity to prevent or repair the damage caused [13,14].

Sharks are among the most threatened marine animals, due to overfishing, the destruction of their habitats, and climate change [19,20,21]. In fact, over the past 50 years, the populations of certain oceanic sharks and rays have declined by more than 70% [21]. Recent studies have also shown that climate-change-related stressors affect the physiology and behavior of sharks [19,22,23]. Predator avoidance represents a critical behavior for survival in the ocean. During early ontogeny, sharks exhibit different strategies to avoid predators. For instance, in oviparous species, embryos interrupt respiratory gill movement and tail beats to become undetectable in a predator’s presence [24]. In the small-spotted catshark *Scyliorhinus canicula,* this behavior is performed when the embryo is beyond stage four of development, when it starts to be directly exposed to the surrounding seawater [25]. Any variable that interferes with the duration of the freezing response increases or reduces the chance of embryo survival [24,26]. *S. canicula* is a small oviparous elasmobranch fish from temperate regions that inhabits the Eastern Atlantic Ocean (i.e., from the coast of Norway to the south of Senegal and the Mediterranean Sea) [27]. It lays its eggs in pairs encased in a protective egg capsule in the shallow sub-littoral zone to the edge of the continental shelf [27], where they are exposed to variations in environmental conditions [28]. In fact, Ripley et al. [26] demonstrated that ocean warming impairs predator avoidance behavior in embryos of *S. canicula*.

Here, we investigate the potential effects of the understudied end-of-the-century projected ocean oxygen content (93% air saturation) and hypoxia (26% air saturation) on the survival, behavior (namely the freezing response duration and tail beat frequency), and oxidative stress of *S. canicula* embryos. The enzymes assessed here are the main biomarkers used to evaluate the response of fish to stressors such as environmental hypoxia, temperature, and pollutants [29]. As they act in sequence, we measured the multiple enzymes that participate in the conversion of ROS into less harmful forms [12,13]. We hypothesized a significant production of ROS (super oxide) and, consequently, a significant increase in super oxide dismutase production in the hypoxia treatment. An experiment-based risk assessment of sharks to climate change can provide managers and policy-makers with information that may assist them to take proactive measures targeting endangered species [19].

## 2. Materials and Methods

### 2.1. Collection and Incubation

In February 2021, *S. canicula* adults (two females and one male) were captured by fishermen at Figueira da Foz (west coast of Portugal, 40°09′21″ N, 8°52′33″ W) and transported under controlled conditions to “Laboratório Marítimo da Guia” facilities (Cascais, Portugal, 38°41′42″ N, 9°27′8″ W). Upon their arrival, sharks were acclimated to the laboratory in semi-closed aquatic life-support systems (LSS; see schematic representation in Supplementary Material). Within the LSS, seawater temperature was maintained at 16.2 °C ± 0.2, and organisms were fed four to six percent of their body weight per week^−1^ in fish and squid [30]. The LSS were equipped with protein skimmers (Schuran, Germany), as well as mechanical and biological filtrations (sand filters FSBF 1500, TMC Iberia, Portugal; Ouriço^®^ bio balls, Fernando Ribeiro, Portugal). The temperature was automatically controlled (T controller twin, Aquamedic, Bissendorf, Germany) and maintained through a centralized seawater chilling system (Frimar, Fernando Ribeiro Lda, Barcarena, Portugal) and submerged seawater heaters (EHEIM, 100 W, Deizisau, Germany). Monitoring of dissolved oxygen (DO 210, VWR, Radnor, PA, USA), pH (pHenomenal^®^ pH 1100 H, VWR, Radnor, PA, USA), and salinity (HI98319, Hanna Instruments, Woonsocket, RI, USA) was daily performed. Total ammonia (NH_3_, NH_4_^+^), nitrite (NO_2_^−^) and nitrate (NO_3_^−^) levels were assessed (Tropic Marin^®^, Wartenberg, Germany) twice. Ammonia and nitrite were kept below detectable levels and nitrate below one mg mL^−1^. Egg deposition was daily monitored, and the paired eggs (the two eggs, one per oviduct, attached by tendrils deposited in each opposition event) were transferred to another LSS before each date of deposition was documented. According to the scale of Musa et al. [25], when the eggs reached the fourth stage of development (between 7 and 10 weeks, being characterized by the opening of all seawater slits exposing the embryo to the seawater conditions), eggs were randomly assigned to the experiment tanks (*n* = 24). Experimental treatments were as follows: (i) Control (CT = 100% oxygen saturation; *n* = 8); (ii) Deoxygenation (DO = 93% oxygen saturation; *n* = 7), and (iii) Hypoxia (HO = 26% oxygen saturation; *n* = 9). Seawater conditions were set at 16 °C and 8.0 pH units for all treatments. Embryos were divided in three replicates per treatment and each replicate had an independent supply of seawater. Each replicate was composed of a 45 L tank (56 × 39 × 28 cm). The three replicates from each treatment were inserted in the respective seawater bath in a semi-closed system equipped with protein skimmers (Reef Skim Pro 400, TMC-Iberia, São Julião do Tojal, Portugal), biological filtration (Ouriço^®^ bio balls, Fernando Ribeiro, Portugal), and automatic temperature controls (STC-3000) and maintained through a centralized seawater chilling system (Frimar, Fernando Ribeiro Lda, Portugal). Seawater abiotic parameters (i.e., dissolved oxygen, salinity, temperature, and pH) were monitored as in the LSS from adults and NH_3_, NH_4_^+^, NO_2_^−^, and NO_3_^−^ levels were assessed every two days (Table 1). Control, deoxygenation, and hypoxia replicates supplying seawater had independent optometers (PyroScience (Graz, Austria) FireStingO_2_, accuracy ± 0.1 mg O_2_ L^−1^) and, consequently, independent oxygen levels. Deoxygenation and hypoxic treatment replicates were supplied with seawater previously injected in each respective cylindrical column with certified nitrogen gas (Air liquide, Algés, Portugal) via solenoid valves controlled by an Arduino Mega controller (Mucha, 2020; hysteresis set at 0.2 mg O_2_ L^−1^). Oxygen levels were stabilized by injection of filtered atmospheric air (via air stones) whenever necessary. Seawater baths and replicates tanks were continuously fed via a seawater pump (35 W; TMC, V2 Power Pump, 2150 L h^−1^) with oxygen-limited seawater from the cylindrical column, and seawater flow was daily manually adjusted through an acrylic flow meter (1–10 L min^−1^ range). Experimental exposure was performed over six days, under a photoperiod of 12 h/12 h L:D (light:dark cycle). The adult sharks used in this experiment were returned to the sea at the end of this experiment.

### 2.2. Biological Response

Mortality was daily and closely monitored during the six days of exposure following Musa et al. [25]’s method. After six days of exposure, embryos were individually observed for one minute, and the number of tail beats was recorded. Each observation was repeated three times to obtain a mean value of tail beats per minute. Regarding freeze response duration, embryos were flicked for three minutes in their respective treatment tanks to mimic a physical disturbance generated by a predator [26]. Then, a mobile device (Samsung, Galaxy M12, 1080p) was placed in front of the experimental tank for video recording the embryo activity. The respective video footage was used to determine the freeze response duration, defined as the period between the end of physical stimuli and the time that any buccal pumping or uncoiling of the tail is registered [26].

### 2.3. Biochemical Analysis

#### 2.3.1. Tissue Processing

At the end of day six, the embryos were removed from the experimental tanks and the capsule were euthanized (immersion into MS222 solution, buffered with sodium bicarbonate at 1:1 ratio), measured (5 cm ± 1.1), weighed (0.56 g ± 0.38), and individually stored as a whole at −80 °C, until further analyses. Subsequently, the frozen samples were removed from the freezer and kept on ice to maintain protein integrity and enzyme activity. In order to homogenize the samples, five mL of phosphate-buffered saline solution (PBS, pH 7.4: NaCl, NaHPO_4_, KH_2_PO_4_ and milli-Q H_2_O) was added to the samples, and the mixture was grinded with the aid of an Ultra-Turrax (Ika, Staufen, Germany). The homogenates were centrifuged (21,000× *g*, 10 min, 4 °C) in microtubes (1.5 mL), after which the supernatant was removed, transferred to a new microtube, and frozen at −80 °C.

#### 2.3.2. Total Protein Content

Bradford [31] protocol was followed to quantify the total protein in each sample. In this assay, bovine serum albumin (BSA) was diluted in PBS to obtain protein standards (from 0.0625 to 4 mg mL^−1^) to obtain a calibration curve. Blanks were made with PBS. Standards or samples (20 µL) and 180 µL Bradford reagent were added to each well in 96-well microplates, in two replicates. Absorbances were read in the spectrophotometer (Biotek Synergy HTX multi-mode reader, Winooski, VT, USA) at 595 nm. Whenever protein concentration exceeded four mg mL^−1^, samples were diluted to 1:2. A calibration curve was constructed with the standards to quantify the total protein in each sample.

#### 2.3.3. Antioxidant Enzymes

##### Superoxide Dismutase Activity (SOD)

Superoxide dismutase activity was measured by following the McCord and Fridovich [32] method, adapted for microplates. Briefly, to each 96-well microplate well, 200 µL of buffer solution (0.5 mmol L^−1^ potassium phosphate, pH ~ 8.0, Sigma-Aldrich, Schnelldorf, Germany), 10 µL of EDTA (3 mmol L^−1^; Riedelde Haën, Seelze, Germany), 10 µL of xanthine (3 mmol L^−1^; Sigma, Schnelldorf, Germany), 10 µL of NBT (0.75 mmol L^−1^; Sigma-Aldrich, Schnelldorf, Germany), and 10 µL of sample were sequentially added. Afterwards, 10 µL of 100 mU xanthine oxidase (XOD; Sigma-Aldrich, Schnelldorf, Germany) was added to initiate the reaction. The absorbance was read at 560 nm every two minutes for 20 min with a microplate reader (Biotek Synergy™ HTX Multi-Mode Reader, Winooski, VT, USA). SOD activity results were determined as percentage of inhibition mg^−1^ total protein.

##### Catalase Activity (CAT)

Catalase (CAT) activity was measured following the adapted protocol first described by Beers and Sizer [33]. Calibration was performed using three mL of Potassium phosphate buffer (K_3_PO_4_; Sigma-Aldrich, Schnelldorf, Germany). Samples (100 µL) and 2.9 mL of hydrogen peroxide (0.036 mol L^−1^ H_2_O_2_; Sigma-Aldrich, Schnelldorf, Germany) were added to 96-well quartz microplates. Absorbance was read at 240 nm every 42 s over four minutes with a microplate reader (Biotek Synergy™ HTX Multi-Mode Reader, Winooski, VT, USA). Catalase activity was calculated using the H_2_O_2_ molar extinction coefficient (0.0436 ε^mmol L−1^). Results were expressed relative to the total protein content (nmol min^−1^ mg^−1^ total protein).

##### Glutathione Peroxidase Activity (GPx)

Glutathione peroxidase activity was determined according to the method described by Lawrence and Burk [34]. Briefly, in a 96-well microplate,120 µL of buffer (50 mmol L^−1^ phosphate with 5 mmol L^−1^ EDTA, pH 7.4), 50 µL of co-substrate solution (0.8 mmol L^−1^ β-NADPH, 4 mmol L^−1^ reduced glutathione, 4 UmL^−1^ glutathione reductase, and 4 mmol L^−1^ sodium azide), and 20 µL of sample were added. To each well, 20 µL of cumene hydroperoxide (C_9_H_12_O_2_) was added to start the reaction, and the microplate was slowly and gently shaken to mix contents. The absorbance was read at 340 nm every minute for 15 min with a microplate reader (Biotek Synergy™ HTX Multi-Mode Reader, Winooski, VT, USA). GPx activity was determined with the β-NADPH coefficient extinction 3.73 ε^mmol L−1^. Results were expressed as nmol min^−1^ mg^−1^ total protein.

##### Glutathione-S-Transferase (GST)

Glutathione S-transferase (GST) activity was determined according to the method described by Habig et al. [35] and adapted for microplates. The substrate solution was prepared with 9.8 mL of Dulbecco’s phosphate buffered saline solution, 0.1 mL of reduced glutathione (200 mmol L^−1^ in Milli-Q water), and 0.1 mL of 1-chloro-2,4-dinitrobenzene (CDNB) (100 mmol L^−1^ in 95% ethanol), all from Sigma-Aldrich (Burlington, MA, USA). This solution was added to 96-well microplates (180 µL) with 20 µL of sample in each well. The absorbances were read at 340 nm every minute for six minutes, using a microplate reader (Biotek Synergy™ HTX Multi-Mode Reader, Winooski, VT, USA). GST activity was calculated using the molar extinction coefficient for CDNB of 5.3 ε^mmol L−1^. Results were expressed as nmol min^−1^ mg^−1^ total protein.

### 2.4. Protein Repair and Elimination

#### 2.4.1. Heat Shock Response (HSP70)

Heat shock protein 70 (HSP70) was quantified through enzyme-linked immunosorbent assay (ELISA), adapted from the method described by Njemini et al. [36]. Standard solutions were prepared from HSP70 active protein (Origene, Rockville, MD, USA) to obtain a curve ranging from 0.0078 to 2 µg mL^−1^. Samples (50 µL) were added to a 96-well high-binding microplate (Greiner, Bio-One, Kremsmünster, Austria) and incubated overnight at 4 °C (covered with aluminum foil). On the following day, the microplate was washed twice using PBS (containing 0.05% TWEEN-20) and a blocking solution (100 µL; 1% BSA (bovine serum albumin, Sigma-Aldrich, Burlington, MA, USA) in PBS) was added. The microplate was covered in aluminum foil and incubated at 4 °C overnight. The microplate was washed thrice with PBS. Afterwards, 50 µL of primary antibody (5 µg mL^−1^ in 1% BSA: anti-HSP70/HSC70, Origene, Rockville, MD, USA) was added to each well. Then, the microplate was covered in aluminum foil and incubated at 4 °C overnight. The next day, the microplate was washed again thrice with PBS (0.05% TWEEN-20) and a secondary antibody conjugated with alkaline phosphatase (anti-mouse IgG Fc specific, Sigma-Aldrich, Schnelldorf, Germany) diluted to 1 µg mL^−1^ was added to each well. Then, the microplate was covered in aluminum foil and incubated at 37 °C for 90 min before being washed again. Alkaline phosphatase substrate solution (100 µL; 100 mmol L^−1^ NaCl; Panreac, Barcelona, Spain), MgCl_2_ (50 mmol L^−1^ Sigma-Aldrich, Schnelldorf, Germany), Tris-HCl (100 mmol L^−1^; Sigma-Aldrich, Schnelldorf, Germany), and PnPP (27 mmol L^−1^; pH 8.5, 4-nitrophenylphosphate disodium salt hexahydrate; Sigma-Aldrich, Schnelldorf, Germany) were added to each microplate well. Subsequently, it was left to incubate for 30 min while covered in aluminum foil in a shaker at room temperature. Finally, 50 µL of stop solution (3 mol L^−1^; NaOH; Panreac, Barcelona, Spain) was added to each microplate well, and the absorbance was read at 405 nm. The results from HSP70 were normalized with the sample protein content and expressed as µg mg^−1^ total protein.

#### 2.4.2. Ubiquitin Content (Ub)

Ubiquitin (Ub) content was determined through an ELISA as described by Lopes et al. [37]. Each sample (100 μL) was added to 96-well microplates (Greiner, Bio-One, Kremsmünster, Austria) and incubated overnight at 4 °C. After 24 h, microplates were washed two times with PBS containing 0.05% TWEEN 20, and 100 μL of blocking solution (1% BSA) was added to each well. The microplates were then incubated for 90 min at 37 °C. Subsequently, 50 μL of primary antibody (P4D1, sc-8017, HRP conjugated; Santa Cruz, CA, USA) was added to each well. After another overnight incubation period at 4 °C, microplates were washed three times to remove non-linked antibodies. Afterwards, 100 μl of substrate (TMB/E, Merck Millipore, St. Louis, MO, USA) was added to each microplate well and let to incubate for about 30 min at room temperature. Then 100 μL of stop solution (1 mol L^−1^; HCl; Panreac, Barcelona, Spain) was added to each well. Absorbances were read at 450 nm, using a microplate reader (Biotek Synergy™ HTX Multi-Mode Reader, Winooski, VT, USA). The Ub content was calculated from the calibration curve, based on serial dilutions of purified ubiquitin (0–1 μg mL^−1^, UbpBio, E-1100, Dallas, TX, USA). Results were expressed as µg mg^−1^ total protein.

#### 2.4.3. Lipid Peroxidation

Lipid peroxidation was determined by determining the malondialdehyde (MDA) content, as a specific end-product of lipid oxidative degradation, based on the thiobarbituric acid substance (TBARS) assay [38] and following essentially the protocol from Ohkawa et al. [36]. Samples were measured after building a calibration curve with MDA (Sigma-Aldrich, Burlington, MA, USA) as standard (0.001 to 0 0.1 µmol L^−1^ MDA prepared in Milli-Q water (MQW)). In 1.5 mL microtubes, to 5 µL of each standard solution or sample, 45 µL of monobasic sodium phosphate buffer (50 mmol L^−1^ in MQW, pH 7–7.4), 12.5 µL sodium dodecyl sulphate (8.1% in MQW), 93.5 µL of trichloroacetic acid (20% in MQW, pH 3.5), 93.5 µL of thiobarbituric acid (1% in MQW, centrifuged, in which a sodium hydroxide pill was dissolved), and 50.5 µL of MQW were added. The mixture was centrifuged for 30 s and incubated in a boiling water bath for 10 min. The mixture was then placed on ice for five minutes. MQW (62.5 µL) was added to each microtube and centrifuged at 2000× *g* for five minutes. Then, 150 µL of each microtube was added to microplate well. Absorbances were read at 532 nm (Biotek Synergy™ HTX Multi-Mode Reader, Winooski, VT, USA).

### 2.5. Statistical Analysis

Linear models (LMs; “glm” function) were used to infer the effects of treatments on the oxidative stress bioindicators. Treatment was set as a three-level factor (Control, Deoxygenation, and Hypoxia, as well as replicates (Systems A, B, and C). The admissible error was set at 0.05 and *p*-values were corrected by Tukey post-hoc tests (“emmeans” function in the “emmeans” package). Models from the Gaussian family (identity link) were fitted to Catalase, MDA, SOD, GPx, GST, HSP70, ubiquitin, and DNA damage levels. To evaluate the effect of the treatments and replicates on the response variables, type II Wald Chi-squared tests (function “Anova”) were performed on the models. The model residuals were plotted to check assumptions of normality, homoscedasticity, and independence among residuals. All statistical analyses were performed using R Studio (v. 2021.09.1).

## 3. Results

The survival rate of *S. canicula* embryos was 100% under the control conditions (CT, Figure 1A). However, under deoxygenation (DO, 93% oxygen saturation) and hypoxia (HO, 26% oxygen saturation), the survival rate decreased to 87.5% and 56%, respectively. Tail beats were significantly higher in the embryos under HO (68.4 ± 6 tail beats min ^−1^; *p* < 0.0001) compared to those under DO (47.8 ± 10 tail beats min ^−1^; *p* < 0.0001) and CT (45.8 ± 10 tail beats min ^−1^; *p* < 0.0001) (Figure 1B). No significant differences in tail beats were observed between the DO and CT conditions (*p* = 0.4879; see detailed statistical information in Table 2 and Table 3).

HO significantly decreased the freeze response duration (1.7 ± 0.5) compared to DO (6.7 ± 3.8; *p* = 0.0113) and CT (7.4 ± 3.2; *p* = 0.0035; Figure 1C). Moreover, there were no significant differences in the embryos’ freeze response duration between the DO and CT conditions (*p* = 0.6491; Figure 1C).

Regarding oxidative stress, no significant effects of the DO and HO conditions were detected on all the biomarkers analyzed (*p* > 0.05). Yet, the defense enzyme machinery (SOD, CAT, GPx, and GST) always showed higher mean values in the HO treatment (Figure 2) compared to those for the CT and DO treatments. Concerning protein repair and removal mechanisms, the HSP, Ub, and MDA mean values were also non-significantly higher under the HO treatment (*p* > 0.05; Figure 3; see detailed statistical information in Table 4).

## 4. Discussion

Dissolved oxygen is an essential element of the marine environment, actively affecting ecological, metabolic, and biogeochemical processes [6,39]. As an immobile stage of shark development, eggs are particularly vulnerable to environmental fluctuations (including oxygen content) in the marine environment [25,28]. Here, we demonstrate that low oxygen content strongly affects the survival of shark embryos. A previous study found a 100% mortality rate for *S. canicula* embryos exposed to 20% oxygen saturation for three weeks [40]. More recently, Musa et al. [28] found a significant decline (63.2% at 15 °C and 61.9% at 20 °C) in the survival rate of *S. canicula* embryos exposed to 50% oxygen saturation. The authors suggested that an increase in the oxygen demand in older and larger embryos is the cause of this hypoxia-induced increase in the late-stage mortality rate.

Tail beats facilitate the entry and exit of water from inside the capsule, allowing a supply of oxygen to the embryo for its development [41]. Exposure to hypoxia triggers a significant difference in tail beat frequency compared to that under the control conditions, suggesting that the embryo responds with an increase in tail beats to maintain its oxygen supply. A similar observation was made by Di Santo et al. [41] with skate embryos, showing an increase in tail beats when exposed to 55% or lower air saturation. Despite its advantage in replenishing oxygen, as a high-energy behavior, an increase in tail beats can lead to a faster depletion of the yolk’s energy reserves before the end of embryonic development. However, Musa and colleagues [28] found no effects on the *S. canicula* yolk consumption rate under 50% oxygen saturation (15 °C), in contrast to those exposed to warming conditions (20 °C). On the other hand, an increase in tail beats can make the embryo more vulnerable, as it releases more chemical and physical signals that can be sensed by predators [24,42]. Moreover, behavioral trials show that embryos kept in hypoxic conditions have a significantly shorter freeze response duration. When in the presence of predators, embryos respond by ceasing their movement to go unnoticed [24,26]. Therefore, any factor that interferes with this behavior can mean the difference between survival and death [24,26]. Here, we demonstrate that low oxygen content affects shark embryos directly by causing death and indirectly by weakening its predator avoidance behavior. In this sense, we hypothesize that adults of oviparous species will avoid areas with low oxygen levels and look for other areas with better conditions to lay their eggs. A previous study also found that future predicted warming also has the same impact on shark embryos by reducing their freeze response duration [26].

SOD, CAT, GPx, and GST are the main antioxidant enzymes and important indicators of oxidative stress [43]. In this study, there were no significant variations in the activity levels of these antioxidant enzymes among the different treatments. The same findings were observed for protein repair mechanisms and cellular damage. *S. canicula* eggs are known have a great ability to deal with variations in oxygen content [40]. As they are deposited in coastal areas experiencing fluctuations in oxygen availability [27,28], the species may have developed adaptations to avoid the occurrence of oxidative stress throughout its evolution. Furthermore, sharks are known to produce a high level of nonenzymatic ROS scavengers involved in the prevention of oxidative damage, such as ascorbic and uric acids, carotenoids, reduced glutathione, and amino acids [37,44,45]. ROS scavengers have a low molecular weight, act as complementary antioxidant defense system, and are prevalent in sharks [44,46]. In a previous study with *S. canicula* juveniles exposed to long-term and high CO_2_ conditions, there was also no evidence of oxidative damage [45]. The same finding was observed for the tropical benthic shark *Chiloscyllium plagiosum* in which their exposure to a high level of CO_2_ for 50 days did not elicit oxidative damage [37]. The authors argued that sharks’ non-enzymatic ancient antioxidant system plays a key role in the defense against oxidative damage. The resiliency of sharks that has allowed them to survive several mass extinction events may be, in part, due to an efficient combination of enzymatic and non-enzymatic defense systems to cope with changes in parameters in the environment. However, the lack of evidence of increased oxidative stress and cellular damage may be due to the short time (six day) of exposure tested in the present study. Therefore, studies with longer exposure are warranted to test this hypothesis. Unlike elasmobranchs, studies have shown that teleost fish are less resilient to low-oxygen conditions [29,47,48]. For example, it was found that hypoxia induces oxidative stress in several tissues of *Perccottus glenii* after a short period of exposition [47].

## 5. Conclusions

Here, we demonstrate that a reduction in the level of oxygen in seawater poses direct and indirect threats to shark embryos, causing an increased mortality rate and leaving them more vulnerable to predators. As the embryos are at a sensitive and critical ontogenetic stage, any interference can constitute a threat to the survival of the species. The projected levels of ocean deoxygenation for the end of the century (seven percent lower) does not represent a major effect to *S. canicula*, as this is a coastal species already experiencing fluctuation in oxygen levels. In contrast, lower oxygen levels cause direct and indirect increased embryo mortality rates. As hypoxic zones are increasing in coastal areas, particularly in areas where this species occurs, we may see a decline in this species in the future. The reduction in the freeze response duration and the increase in the tail beat rate evidenced in this study can be even more aggravated when we consider the synergistic effects of other environmental changes, such as the increase in temperature, for example. Therefore, future studies addressing the interplay between hypoxia and other climate-change-related drivers (e.g., warming and acidification) are greatly needed (see also [4,5]) to better understand how sharks may fare in the ocean of tomorrow.

## Figures and Tables

**Figure 1 biology-12-00577-f001:**
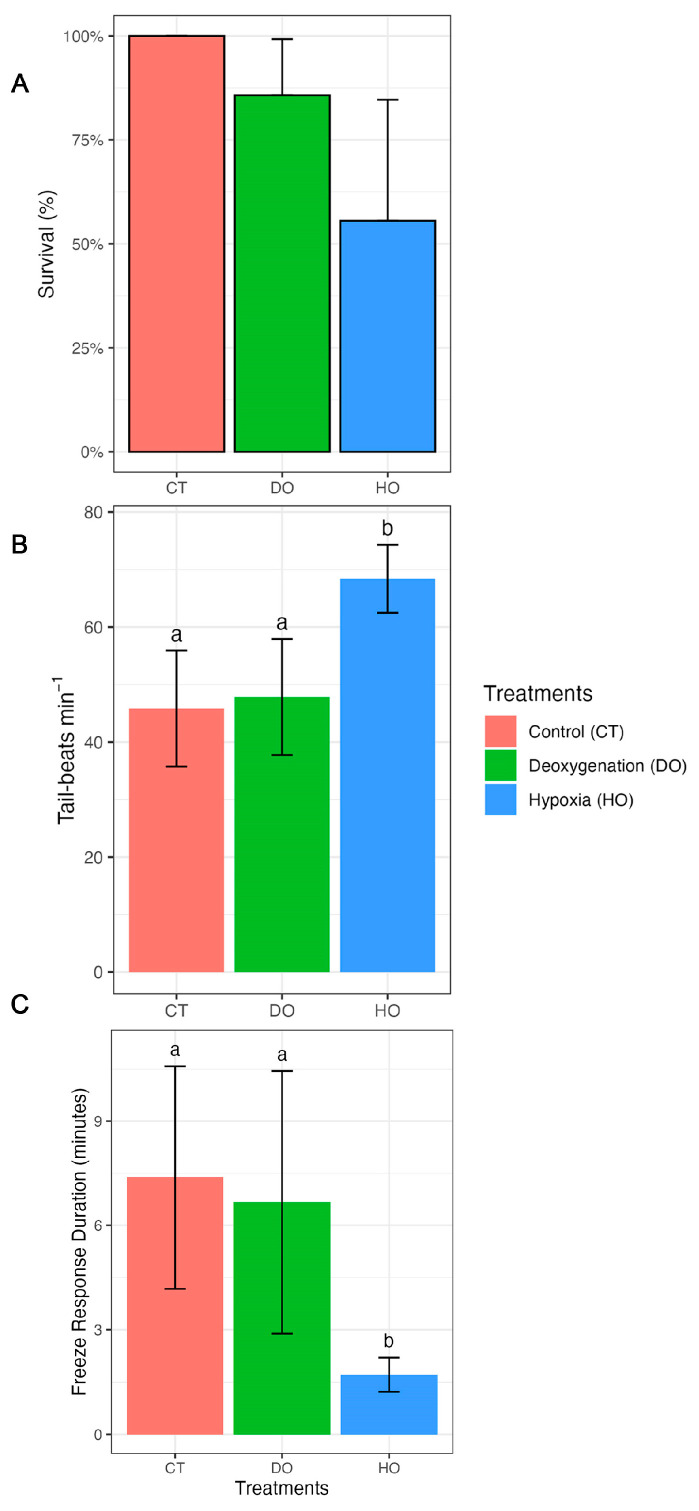
Effect of control (100% air saturation), deoxygenation (DO; 93% air saturation), and hypoxia (HO; 26% air saturation) on survival (**A**), tail beat rate (**B**), and freeze response duration (**C**) of *S. canicula* embryos. Different lower-case letters indicate statistically significant differences between experimental treatments.

**Figure 2 biology-12-00577-f002:**
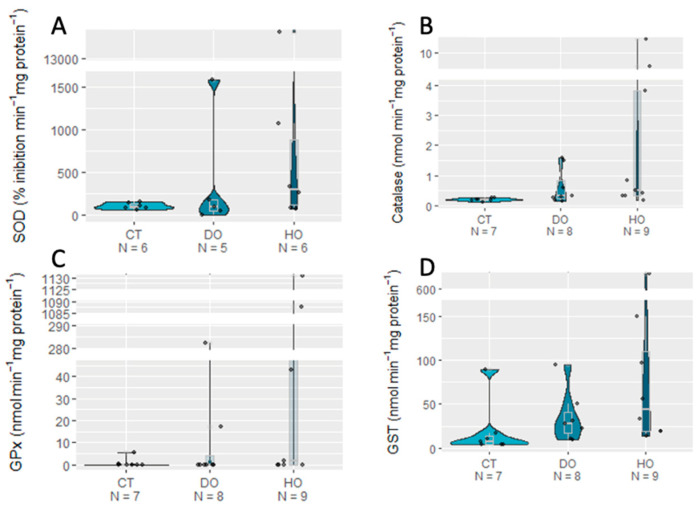
Antioxidant enzyme activities SOD (**A**), CAT (**B**), GPx (**C**), and GST (**D**) of *S. canicula* embryos exposed to experimental conditions (CT _ Control, DO _ Deoxygenation, and HO _ Hypoxia).

**Figure 3 biology-12-00577-f003:**
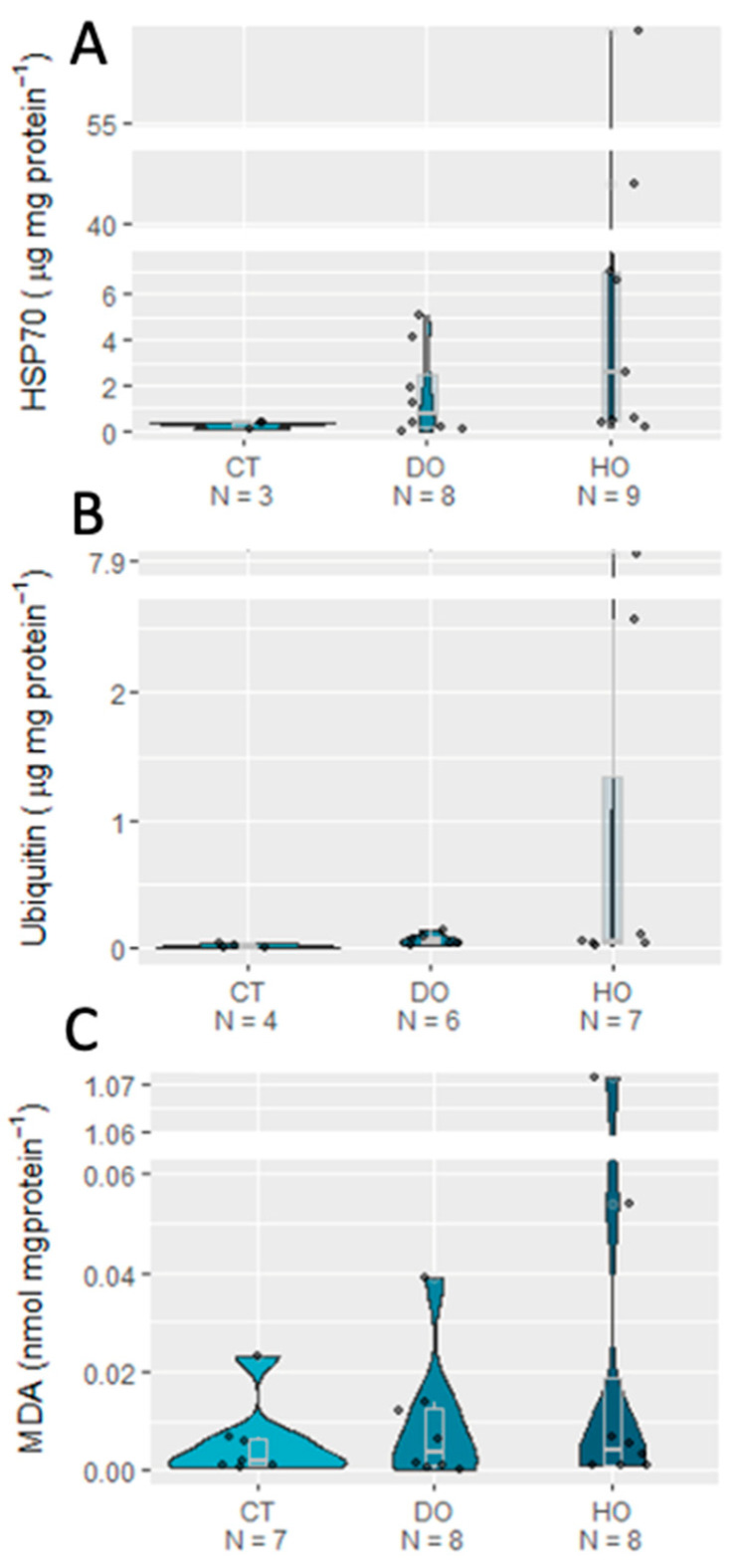
Effects of control (100% air saturation), deoxygenation (93% air saturation), and hypoxia (26% air saturation) on protein repair and elimination (HSP70 (**A**), Ub (**B**), and MDA (**C**)) of *S. canicula* embryos exposed to experimental treatments (CT _ Control, DO _ Deoxygenation, and HO _ Hypoxia).

**Table 1 biology-12-00577-t001:** Seawater parameters measured during the experiment. Values are represented as means ± standard deviation.

	Control	Deoxygenation	Hypoxia
Temperature (°C)	16.3 ± 0.2	16.1 ± 0.2	16.3 ± 0.1
pH (pH units)	8.07 ± 0.02	8.07 ± 0.02	8.16 ± 0.02
Salinity (g/L)	33.0 ± 0.2	33.0 ± 0.2	33.0 ± 0.2
Oxygen (%)	99.56 ± 0.69	92.83 ± 0.93	28.26 ± 3.25

**Table 2 biology-12-00577-t002:** Summary output for generalized linear models of shark embryos’ survival, average number of tail beats, and freeze response duration among control, deoxygenation, and hypoxia treatments. Statistical significance at *p*-value < 0.05 is shown in bold.

	Estimate	SE	Df	z-Ratio	*p*-Value
Survival					
CT–DO	23.13	55,398.03	NA	0.000	1.0000
CT–HO	24.70	55,398.03	NA	0.000	1.0000
DO–HO	1.57	1.27	NA	1.234	0.6520
	**Estimate**	**SE**	**Df**	**t-Ratio**	***p*-Value**
Tail beats					
CT–DO	−2.0	2.86	54	−0.698	0.4879
CT–HO	−22.6	3.02	54	−7.465	**<0.0001**
DO–HO	−20.6	3.21	54	−6.405	**<0.0001**
Freeze response duration					
CT–DO	0.708	1.53	18	0.463	0.6491
CT–HO	5.661	1.47	18	3.858	**0.0035**
DO–HO	4.952	1.58	18	3.140	**0.0113**

**Table 3 biology-12-00577-t003:** Analysis of deviance table (Type II tests) for the generalized mixed models for survival, average number of tail beats, and freeze response duration of shark embryos exposed to control, deoxygenation, and hypoxia. Values of *p* < 0.05 are shown in bold.

	Chi-Squared	Df	*p*-Value
Survival	6.4566	2	**0.03963**
	**Sum of Squares**	**Df**	***p*-Value**
Tail beats	5250.3	2	**1.024 × 10^−9^**
Freeze response duration	135.65	2	**0.002595**

**Table 4 biology-12-00577-t004:** Analysis of deviance table (Type II tests) for the generalized mixed models for oxidative stress biomarker analysis of shark embryos exposed to control, deoxygenation, and hypoxia.

	Chi-Squared	Df	*p*-Value
SOD	2.0314	2	0.3621
CAT	5.3528	2	0.06881
GPx	3.5299	2	0.1712
GST	3.1165	2	0.2105
HSP70	3.1764	2	0.2043
Ub	2.4077	2	0.3
MDA	1.9324	2	0.3805
DNA Damage	0.68601	2	0.7096

## Data Availability

The data available upon request to the corresponding author.

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
