# Peer review of "Impacts of Deoxygenation and Hypoxia on Shark Embryos Anti-Predator Behavior and Oxidative Stress"

_biology, 2023, doi:10.3390/biology12040577_

Round 1

Reviewer 1 Report

Very interesting study and of paramount importance in the current climate change scenario. Minor details only to correct.

1.     How long were the adult sharks acclimated for? How did you know they would put eggs? Were they weighed and measured? How did you know they were adults, size? (of course, they later laid eggs, but prior to this?). 

2.     Do you have capture licenses or permits? Please insert.

3.     What do you mean by “paired eggs”? Were they measured?

4.     Line 116 - 1 mgmL-1, correct

5.     Line 123: numbers from 1-9 should be rewritten in full, and >10 are numerical. So, change “3 replicates’ to “three replicates”. Same in line 121, two days instead of 2, line 50, six days of exposure.

6.     Several O2 require the 2 positioning to be corrected to subscript throughout the manuscript.

7.     What did you do with the adults after oviposition?

8.     You homogenized the entire embryos, and did not dissect them and remove specific organs, is this correct? This is not clear in the manuscript, please indicate. Why did you do so? Were the embryos measured and/or sexed?

9.     Correct units to IUPAC standards throughout the entire text, mM should read mmol L-1, M should read mol L-1, etc.

10.  The Enzyme-Linked Immunosorbent As- 226 say (ELISA) section is not clear, you produced the ELISA in the lab, in microplates? Or did you buy the kit commercially? Where did you get the HSP70 solution? Commercial? And the 96-well high-binding microplate, what brand and type?

11.  Line 223 - 50μL - Correct to 50 μL (space between number and numeral), also required for line 238 (37 oC), line 272 – 30s, which should read 30 s… verify throughout and insert spaced before units.

12.  Have other studies been conducted concerning deoxygenation an hypoxia in sharks and oxidative stress parameters analyzed? Please insert.

13.  Figure 1 (A) – error bars are missing below, they should be plotted as B and C. 

14.  I think your conclusions should be a little “fleshed out”, you did not detail the findings very much. Which deleterious effects, for example?

15.  Line 387 – SPECIES, please! Like “lápis” in Portuguese, there is always an S at the end, even if singular 

16.  Data Availability Statement is the MDPI standard, please verify.

Reviewer 2 Report

I thought this was a very interesting study on the interaction of (probably the commonest European) shark with changing oxygen levels. It is worthy of publication as it is, but I feel that would be wrong because there are a lot of important implications for this study that are not mentioned.

There should have been a bit more near the start about introducing the use of antioxidant enzymes- why were these chosen, what was the hypothesis of what they might show, why multiple examples used? I think these all beed to be answered.

As far as the implications go, some may not be able to answer, some you may be able to hypothesise, some you may have a clear answer:

How might this be expected to impact on the relative proportions of egg laying and livebearing sharks in areas of low oxygen? Presumably the former swim away.

Is there any indication that Scyliorhinus populations have changed in areas of local anoxia, such as within the Adriatic?

A large proportion of benthic chondrichthyes in deep, oxygen poor water water are egg layers (scyliorhinids such as Apristurus, rajids, chimaeroids). Could eggs be better at surviving very low oxygen events that embryos within an adult?

Could this be linked to chondrichthyes often being common near oxygen minimum zones?

At least one major extinction event (Cenomanian-Turonian) was linked to widespread anoxia, and scyliorhinids were diverse at the time. Does this study lend any information to interpreting extinction events?

Marine chondrichthyans seem to have been amongst the least impacted of groups in anoxia-related extinction events (end Devonian, end Permian, end Triassic). Can this study help explain that?

Reviewer 3 Report

The manuscript entitled "Impacts of deoxygenation and hypoxia on shark embryos anti-predator behavior and oxidative stress" by Varela et al. is an interesting of the impact of deoxygenation (93% air saturation) and hypoxia (26% air saturation) on the anti-predatory behavior and physiology of shark embryos.

In my opinion, I consider this contribution a well-written article on a significant subject, especially with regards to climate change.

The introduction, material, and methods, as well as the discussion are well explained and provide all the necessary information.

For these reason I think this manuscript will be of great interest for the readership of "Biology".

I have few suggestions, listed hereafter:

1- Clarify this statement in the summary, please be more specific: "Among the several climate change-related drivers of change, deoxygenation (oxygen loss) is perhaps the less-studied one"

2- In the abstract: "Climate change is leading to the loss of oxygen content in the oceans and endangering the survival of many marine species", how climate change is leading to the loss of oxygen? Please explain.

3- Can you be more precise in the Material and methods: how many embryos were tested? Only three?

4- Some parts missing in figure 2

5- Conclusion: "The projected levels of ocean deoxygenation for the end of the century had no major effects on the fitness of S. canicula embryos. Yet, hypoxia elicited significant deleterious effects." How do you explain that?

6- Can you better report and explain in the discussion how the analyses of key biomarkers do not show evidence of increased oxidative stress and cell damage under hypoxia.
